# The Dystrophin Node as Integrator of Cytoskeletal Organization, Lateral Force Transmission, Fiber Stability and Cellular Signaling in Skeletal Muscle

**DOI:** 10.3390/proteomes9010009

**Published:** 2021-02-02

**Authors:** Paul Dowling, Stephen Gargan, Sandra Murphy, Margit Zweyer, Hemmen Sabir, Dieter Swandulla, Kay Ohlendieck

**Affiliations:** 1Department of Biology, Maynooth University, National University of Ireland, W23F2H6 Maynooth, Co. Kildare, Ireland; paul.dowling@mu.ie (P.D.); stephen.gargan@mu.ie (S.G.); 2Kathleen Lonsdale Institute for Human Health Research, Maynooth University, W23F2H6 Maynooth, Co. Kildare, Ireland; 3Newcastle Fibrosis Research Group, Newcastle University Biosciences Institute, Faculty of Medical Sciences, Newcastle University, Newcastle upon Tyne NE24HH, UK; sandra.murphy@newcastle.ac.uk; 4Department of Neonatology and Paediatric Intensive Care, Children’s Hospital, University of Bonn, D53113 Bonn, Germany; margit.zweyer@dzne.de (M.Z.); hemmen.sabir@dzne.de (H.S.); 5Institute of Physiology II, University of Bonn, D53115 Bonn, Germany; dieter.swandulla@ukbonn.de

**Keywords:** cytoskeleton, dystrophin, dystrophin–glycoprotein complex, dystroglycan, sarcolemma

## Abstract

The systematic bioanalytical characterization of the protein product of the *DMD* gene, which is defective in the pediatric disorder Duchenne muscular dystrophy, led to the discovery of the membrane cytoskeletal protein dystrophin. Its full-length muscle isoform Dp427-M is tightly linked to a sarcolemma-associated complex consisting of dystroglycans, sarcoglyans, sarcospan, dystrobrevins and syntrophins. Besides these core members of the dystrophin–glycoprotein complex, the wider dystrophin-associated network includes key proteins belonging to the intracellular cytoskeleton and microtubular assembly, the basal lamina and extracellular matrix, various plasma membrane proteins and cytosolic components. Here, we review the central role of the dystrophin complex as a master node in muscle fibers that integrates cytoskeletal organization and cellular signaling at the muscle periphery, as well as providing sarcolemmal stabilization and contractile force transmission to the extracellular region. The combination of optimized tissue extraction, subcellular fractionation, advanced protein co-purification strategies, immunoprecipitation, liquid chromatography and two-dimensional gel electrophoresis with modern mass spectrometry-based proteomics has confirmed the composition of the core dystrophin complex at the sarcolemma membrane. Importantly, these biochemical and mass spectrometric surveys have identified additional members of the wider dystrophin network including biglycan, cavin, synemin, desmoglein, tubulin, plakoglobin, cytokeratin and a variety of signaling proteins and ion channels.

## 1. Introduction

Following the identification of one of the largest genes in the human genome, the *DMD gene* [1], and the initial characterization of its full-length protein product [2,3], it became clear that dystrophin exists in a variety of isoforms ranging in molecular mass from approximately 71 to 427 kDa [4]. Dystrophins are widely distributed in the body, including voluntary striated muscles, the heart, the brain and various other organs systems [5]. Since the large Dp427-M isoform was shown to be tightly associated with several glycoproteins in skeletal muscle fibers [6,7,8,9,10,11], it was concluded that dystrophin probably functions as a molecular anchoring system in the subsarcolemmal cytoskeleton [12,13]. Comprehensive subcellular fractionation studies in combination with both immunoblotting and mass spectrometry have established that dystrophin is highly enriched in the sarcolemma fraction from skeletal muscle [14,15]. Cell biological studies using immunofluorescence microscopy and immunoelectron microscopy agree with these findings and have localized dystrophin to the cytoplasmic site of the plasmalemma in skeletal muscle fibers [16,17]. Subsequent biochemical and cell biological studies confirmed that the dystrophin–glycoprotein complex provides a stabilizing linkage between the intracellular actin cytoskeleton and laminin-211 in the basal lamina of the extracellular matrix [11,18,19]. Modern mass spectrometry-based proteomic approaches have confirmed the composition of the core complex that associates with dystrophin and dystroglycans at the sarcolemma membrane [14,20,21,22,23,24].

The almost complete loss of dystrophin isoform Dp427-M and concomitant reduction in dystrophin-associated proteins triggers the complex pathophysiology of Duchenne muscular dystrophy [7,25], an X-linked and multifaceted disorder of early childhood that primarily affects striated muscles [26], causing respiratory dysfunction, late-onset cardiomyopathy and scoliosis. Functional disturbances of the nervous system [27] and abnormal energy metabolism are also distinctive features of dystrophinopathy [28,29]. The systematic usage of genetic animal models with dystrophic symptoms was instrumental for the elucidation of the body-wide effects of dystrophin deficiency [30,31,32,33]. Skeletal muscle abnormalities are characterized by a complex molecular and cellular pathogenesis, encompassing complex changes in contractile fibers and their supporting connective tissue and neuronal cells [34]. Cycles of degeneration and regeneration, impaired innervation patterns, progressive necrosis, fatty tissue replacement, reactive myofibrosis and sterile inflammation were established as secondary consequences of a primary defect in the dystrophin gene [35,36,37].

In addition to the involvement of sarcolemmal abnormalities in highly progressive Duchenne muscular dystrophy and more benign forms of Becker muscular dystrophy, changes in the dystrophin complex are also involved in sarcoglycanopathies and alpha-dystroglycanopathies. Sarcoglycanopathies represent subtypes of limb–girdle muscular dystrophy that are caused by primary abnormalities in the dystrophin-associated sarcoglycans and alpha-dystroglycanopathies are associated with the abnormal glycosylation of alpha-dystroglycan [38]. Caveolinopathies are another group of disorders that affect the plasma membrane via alterations in caveolae invaginations. Primary abnormalities in caveolin-3 are associated with certain forms of muscular dystrophy and the altered expression of this protein is postulated to also play a pathophysiological role in dystrophinopathy [39]. Importantly, new findings on the underlying mechanisms of dystrophic alterations are crucial for the identification of robust and disease-specific biomarker molecules [40,41,42,43] and the development of novel diagnostic approaches [44], as well as the design of innovative therapies to restore the dystrophin complex and counteract secondary aspects involved in progressive muscle wasting [45,46,47,48,49].

In this review, an overview of the core members of the dystrophin–glycoprotein complex in normal skeletal muscle is provided, including dystroglycans, sarcoglyans, sarcospan, dystrobrevins and syntrophins. This article then focuses on recent biochemical, proteomic and cell biological investigations aimed at the systematic identification of indirectly associated proteins belonging to the wider dystrophin network. This includes crucial members of the extracellular matrix such as biglycan, the surface desmoglein complex, and members of the cytoskeletal network including vimentin, tubulin, desmin and cytokeratin, as well as the cavin–caveolin complex, plakoglobin, ion channels and various signaling proteins. These newer findings have modified the initial view of dystrophin being a purely structural component that functions as a molecular anchor and shock absorber in the membrane cytoskeleton. The Dp427-M isoform of skeletal muscle acts probably as a crucial integrator at the fiber periphery and is, in conjunction with the integrin–laminin axis, majorly involved in lateral force transmission. In addition, the dystrophin complex has been implicated to provide a master node for cytoskeletal organization and cellular signaling events, which are characterized by the linkage of dystrophin to ion channels, the insulin signaling pathway, nitric oxide-based regulatory processes, kinase signaling pathways and excitation–contraction coupling [36].

## 2. The Core Dystrophin Complex in Skeletal Muscle

The full-length dystrophin isoform Dp427-M belongs to the class of giant muscle proteins [50] and consists of several distinct molecular domains as illustrated in the upper panel of Figure 1. This includes amino-terminal and central actin-binding domains, proline-rich hinge regions, spectrin-like rod domains and crucial carboxy-terminal binding sites for interactions with plasmalemmal and cytosolic components [51,52,53,54]. Dystrophin closely interacts with the integral proteins beta-dystroglycan, alpha/beta/gamma/delta-sarcoglyan and sarcospan of the sarcolemma, the extracellular receptor alpha-dystroglycan and laminin-211, the cytosolic components alpha/beta-dystrobrevin and alpha/beta-syntrophin, and the cortical actin cytoskeleton [9,10,11,12,13], as shown in the lower panel of Figure 1.

Sedimentation analysis of the isolated dystrophin complex suggests a monomeric structure with an apparent molecular mass of 1.2 MDa [55]. In Duchenne muscular dystrophy, alterations in the expression of members of the dystrophin network are closely related to key pathophysiological features in dystrophin-deficient muscles, including degeneration-regeneration cycles, progressive fiber degeneration, fibrosis and chronic inflammation. Detailed reviews have been published on the composition of the core dystrophin complex [19,56,57,58], as well as the role of dystrophin and its associated glycoprotein complex in the multisystemic complications of dystrophinopathy and pathophysiological crosstalk throughout the body [34,35,36,37,59]. Therefore, this article does not attempt to recapitulate these biochemical and pathobiochemical issues in detail, but refers instead to specific aspects that are crucial for our general understanding of the wider functional role of the dystrophin complexome in normal skeletal muscle tissue.

### 2.1. The Dystrophin Node in Skeletal Muscle

A model of the spatial configuration of the core dystrophin complex and its association with the extracellular matrix on the one hand and the intracellular cytoskeletal network of contractile fibers on the other hand is presented in the lower panel of above Figure 1. Specific aspects of dystrophin interaction patterns are discussed in detail in below sections. The cell biological concept that the dystrophin–glycoprotein complex occupies a central position at the fiber periphery is summarized in Figure 2. The diagram shows that the dystrophin-associated surface complex forms an organizing node that is majorly involved in (i) the provision of sarcolemmal membrane integrity via a stabilizing linkage between the intracellular actin cytoskeleton and the extracellular matrix protein laminin [11,13], (ii) the establishment of a molecular scaffold and anchoring system for ion channels and enzymes to mediate cellular signaling processes [60,61] (iii) the organization of actin filament attachment and its associated cytoskeletal network [62,63], and (iv) the mediation of lateral force transmission from sarcomeric contraction to the endomysium and its connected layers of the extracellular matrix [64,65].

### 2.2. The Sarcolemmal Dystrophin Complex and Lateral Force Transmission

The peripheral structure of skeletal muscle fibers functions as an essential physical barrier with its protective basal lamina. The underlying sarcolemma membrane provides the physiological structure for the efficient exchange of ions, metabolites and signaling molecules within the contractile system [66]. The plasmalemma is connected to the terminal cisternae region of the sarcoplasmic reticulum at the triad junctions via its invaginations, the transverse tubules. This intricate membrane assembly and its associated Ca^2+^-handling apparatus is involved in the fine regulation of excitation–contraction coupling, muscle relaxation and ion homeostasis, and encounters enormous physical strain during contraction–relaxation cycles [67]. The dystrophin-associated complex is implicated to act as a biomolecular shock absorber by linking the basal lamina to the actin cytoskeleton and thus preventing rupturing of this muscle membrane system [68,69,70].

At specialized costamere regions, which play both a mechanical and a signaling role, the dystrophin complex forms in conjunction with the integrin–vinculin–talin axis a link to the contractile sarcomere units [63,71]. This bridging structure is postulated to provide an indirect means of lateral force transmission to the collagen-rich muscle exterior, in addition to longitudinal forces that are transmitted directly from the contractile apparatus through the cytosol to the myotendinous junction [72,73,74]. In skeletal muscle fibers, the characteristic longitudinal pattern of A bands and I bands reflect the organization of myosin-containing thick filaments and actin-containing thin filaments with their contractile sarcomeric units, which are positioned between Z discs. Following the energy-dependent crossbridge coupling between myosin heads and actin filaments, thin filaments slide past thick filaments. The force generated by this sarcomeric shortening event is partially transmitted by a lateral direct force between Z-disk structures and the M-line regions of neighboring myofibrils. Costamere structures at the fiber periphery play a central role as sensors of the relative mechanical load and support force transduction across the muscle plasma membrane. Contractile force is then further transmitted to the complex layers of the extracellular matrix, consisting of endomysium, perimysium and epimysium, towards the tendon and bone structure [65,72,73]. The second type of force transmission mechanism works by longitudinal means through internal muscle structures embedded in the cytosol. Both lateral and longitudinal coupling mechanisms act in parallel and ultimately transmit the force generated by the actomyosin apparatus in the sarcomere to the tendon and anchoring structures such as bone, as diagrammatically summarized in the lower panel of Figure 2. The dystrophin-associated dystroglycan subcomplex was shown to play a critical role in the sarcomeric cytoskeleton by limiting contraction-induced injury to skeletal muscle fibers [70].

The elucidation of the multifaceted functions of the dystrophin–glycoprotein complex in maintaining membrane stability during excitation–contraction–relaxation cycles, assisting lateral force transmission through costameres and providing a scaffold for anchoring surface receptors and maintaining cellular signaling mechanisms was carried out by multidisciplinary approaches. This included molecular genetics, biochemical purification strategies, structural/biophysical analysis, mass spectrometric proteomics analysis, bioinformatics, chemical crosslinking, cell biological characterization and comparative biomedical studies. In below sections, information on the findings of these bioanalyses is provided in relation to the individual members of the dystrophin–glycoprotein complex and its wider network of skeletal muscle proteins.

### 2.3. Muscle Dystrophin Dp427-M and Its Associated Glycoprotein Complex

The large muscle isoform of dystrophin is a rod-shaped protein [3] with considerable homology to the actinin superfamily of actin crosslinking components, which also includes utrophin and spectrin [75]. Both, dystrophin isoform Dp427-M of the sarcolemma and its autosomal homologue, utrophin isoform Up395-M of the neuromuscular junction, exhibit typical biochemical properties of cytoskeletal components, such as insolubility in non-ionic detergent and efficient extraction by alkaline treatment [76,77]. Compared to the main components of the contractile actomyosin apparatus and its regulatory sarcomeric elements, dystrophin represents a relatively minor component of the skeletal muscle fiber proteome. However, dystrophin isoform Dp427-M constitutes a considerable fraction of the subsarcolemmal cytoskeleton in contractile tissue [76]. This makes full-length dystrophin an important structural and functional component of the sarcolemmal lattice and costamere structures [71]. Besides being present in contractile fibers, dystrophin isoforms also exist in many non-muscle cells [78]. The various dystrophins are encoded by the 79 exon-spanning *DMD* gene, whereby seven different promoters drive the tissue-specific expression of the full-length isoforms Dp427-B in brain, Dp427-M in muscle and Dp427-P in Purkinje cells [4], as well as the shorter isoforms Dp260-R in retina [79], Dp140-B/K in brain and kidney [80], Dp116-S in Schwann cells [81] and Dp71-G in the brain [82] and a variety of other tissues including the spleen [83]. The promoter for Dp71 also produces the shortest known dystrophin isoform named Dp45, which is located in the central nervous system [84]. Of note, the central nervous system displays one of the greatest varieties of dystrophin isoforms, which are involved in synaptic modulation, neuronal excitability and signal integration. Brain Dp427-B is present in neurons of the cerebral cortex and in cerebellar Purkinje cells, Dp140-B is highly expressed during brain development and Dp71-G is located in both neurons and glia cells in the dentate gyrus [82]. Cognitive impairments and emotional disturbances in Duchenne patients are probably linked to altered dystrophin expression in the central nervous system and this is reflected by structural brain abnormalities [85]. The formation of dystrophin complexes and their involvement in dystrophinopathy-associated brain defects has been reviewed by Waite et al. [86].

The composition of the dystrophin–glycoprotein complex has been extensively investigated using a combination of digitonin-based solubilization, wheat germ agglutinin lectin chromatography, ion exchange chromatography and density gradient ultracentrifugation [7,14,55,87], as well as various chemical crosslinking and immunoprecipitation approaches [8,21,22,23,88,89]. Differential detergent extraction procedures [90], two-dimensional gel electrophoresis [10] or alkaline dissociation [87] can be used to isolate individual dystrophin subcomplexes or separate the dystrophin-associated glycoprotein complex from homogeneous dystrophin molecules. Based on these analyses, the core members of the dystrophin-associated complex can be divided into (i) cytosolic components alpha/beta-dystrobrevin [91,92] and alpha/beta-syntrophin [93,94] that interact with the cysteine-rich domain of dystrophin; (ii) integral glycoproteins, including the alpha/beta/gamma/delta-sarcoglyan subcomplex [95,96,97], the highly hydrophobic protein sarcospan [98,99,100] and the main carboxy-terminal dystrophin-binding partner beta-dystroglycan [101]; (iii) laminin-211 [102] and its extracellular receptor alpha-dystroglycan [103], which is a proteolytic cleavage product of the pre-dystroglycan molecule [104]; and (iv) the intracellular actin cytoskeleton that links to an amino-terminal and a rod domain site of full-length dystrophin [51,55,105].

As reviewed by Tarakci and Berger [97], the sarcoglycan subcomplex is initially assembled by the formation of a core between beta-sarcoglycan and delta-sarcoglycan, which subsequently recruits the other two sarcoglycans. Through interactions with sarcospan and additional dystrophin-associated proteins, the sarcoglycan complex secures the formation and mechanical maintenance of the sarcolemmal dystrophin complex. Besides its integrating role in membrane stabilization, the sarcoglycan subcomplex can be chemically modified during fiber contraction, which provides the transduction of information on relative contractile force into cellular signaling [97]. Interestingly, both components of the dystroglycan subcomplex are products of the same gene, *DAG1*, which encodes a pre-pro-protein version of alpha/beta-dystroglycan that includes a signaling peptide and both subunits [101,106]. The precursor protein is extensively modified by N- and O-glycosylation and undergoes proteolytic processing that generates the integral glycoprotein beta-dystroglycan and the extracellular laminin-binding receptor alpha-dystroglycan [103]. Thus, the two dystroglycans form the backbone of the trans-sarcolemmal linkage between the basal lamina component laminin-211 and the dystrophin-associated actin cytoskeleton in the subsarcolemmal region of skeletal muscle [11,13]. The phosphorylation of beta-dystroglycan, especially intracellular tyrosine residues [107], is a crucial step during interactions with signaling proteins [108]. The phosphorylation of the cysteine-rich region in the carboxy-terminal domain of dystrophin also plays a key role in strengthening the interaction with beta-dystroglycan [109]. Thus, post-translational modifications are important modulators of dynamic associations within the dystrophin–dystroglycan axis.

## 3. Proteomic and Biochemical Characterization of the Dystrophin Network in Skeletal Muscle

Over the last two decades, biochemical analyses and mass spectrometry-based proteomics were instrumental to confirm the composition of the dystrophin–glycoprotein complex. Based on these findings, proteomic approaches were extremely helpful for the subsequent identification of novel binding proteins that belong to the wider dystrophin complexome.

### 3.1. Proteomics of the Dystrophin Complex from Skeletal Muscle

Following the discovery and initial biochemical and cell biological characterization of dystrophin, a variety of large-scale screening analyses were carried out to identify proteins that exhibit a close linkage to this membrane cytoskeletal component. Figure 3 illustrates how proteomic surveys and the usage of advanced mass spectrometry has helped to refine this search for novel dystrophin-binding partners and related proteins.

Biochemical and proteomic screening studies have corroborated the direct binding partners of dystrophin, i.e., the integral glycoprotein beta-dystroglycan of the sarcolemma and the cytosolic components dystrobrevin, syntrophin and cortical actin [20,21,22,23,24,110,111]. The other main components of the core dystrophin complex were also identified by proteomics, including the sarcoglycans of the plasma membrane, the highly hydrophobic protein sarcospan and laminin-211 of the basal lamina [21,23]. In addition, comprehensive mass spectrometric studies identified further interaction patterns based on indirect associations with dystrophin isoform Dp427-M [21,22,23,24,89]. These new components of the wider dystrophin network include the extracellular matrix components collagen, fibronectin and biglycan, the plasmalemma proteins integrin, cavin and caveolin, and the cytoskeletal proteins cytokeratin, synemin, actinin, desmin, plectin, desmoglein, desmoplakin and tubulin.

When used in an optimized way, the chemical crosslinking technique can be utilized for the stabilization of fragile protein complexes that would otherwise disintegrate during elaborate subcellular fractionation and extensive biochemical isolation procedures. In protein biochemistry, chemical crosslinking is defined as the intra- and/or inter-molecular stabilization of protein molecules via covalent bonding. Since the chemical crosslinking technique is capable of determining molecular changes in macromolecular oligomerization, it is an ideal bioanalytical tool for studying dynamic protein interaction patterns. The select joining of two or more protein species is achieved by the incubation of protein mixtures with a large variety of crosslinking agents that differ in their molecular spacer arm length, their solubility and chemical reactivity. Frequently used crosslinkers contain reactive groups that are suitable for optimum interactions with functional groups in protein molecules such as primary amines. For example, the crosslinker used in below Figure 4 is a homo-bifunctional, water-soluble, non-cleavable and amine-reactive agent that is highly useful for joining proteins in complex assemblies under physiological conditions. Following the addition of crosslinking agents, protein samples are usually incubated for 30 min at 25 °C for 30 min and then the reaction quenched by the addition of ammonium acetate. A convenient way to analyze the occurrence of crosslinked protein complexes is the determination of an altered electrophoretic mobility in one- or two-dimensional gels [89]. A crucial issue with this method is the potential occurrence of random crosslinking events. However, this can be avoided by employing a low ratio of crosslinking agent to protein and using proper conditions in relation to length of incubation time, efficient quenching of the crosslinking reaction, temperature, buffering and pH-value [112]. Chemical crosslinking has been established as an excellent method in proteomics for studying sensitive protein–protein interactions [113]. Since a large collection of crosslinking molecules exist that greatly vary in their specific spacer arm length, experiments with different agents can be designed to determine spatial constraints within proteins. If a chemical crosslinker with an extremely short spacer arm length is employed, this approach can even stabilize protein interactions within biological membranes that exist in extremely close proximity to another [114].

One of the earliest biochemical studies on the identification of dystrophin-associated proteins used chemical crosslinking [8] and these findings were confirmed by combining protein crosslinking with immunoblotting [88]. The combination of chemical crosslinking and mass spectrometric analysis established that a subpopulation of the sarcolemmal protein cavin-1 exists in a surface complex with beta-dystroglycan in skeletal muscle [89], which has previously been demonstrated in cardiac fibers [21]. Cavins are adapter proteins and mediate in conjunction with the structural caveolin proteins the formation and organization of small invaginations of the muscle plasma membrane, the bulb-like membrane pits called caveolae [115]. These structures are involved in lipid storage, endocytosis, cellular signaling and mechano-protection [116]. Freeze-fracture electron microscopy has established that the number and structure of caveolae are changed in dystrophinopathy, suggesting an important role of abnormal cavins in muscular dystrophy [39,117]. Of note, the crucial repair protein myoferlin [118] is also closely associated with caveolin and intrinsically involved in the maintenance and structural integrity of the sarcolemmal membrane, together with the Ca^2+^-dependent repair protein dysferlin [119].

Figure 4 outlines how a chemical crosslinking mass spectrometry (XL-MS) approach can be employed to study novel protein–protein interaction patterns, which are indirectly linked to dystrophin expression levels [112]. A variety of highly useful bioinformatic programs are available for the visualization of potential protein–protein interactions [120], as shown in the below images, which has been carried out with the STRING program [121]. In recent years, a variety of advances have occurred in XL-MS methodology, such as the development of MS-cleavable crosslinkers and software programs (XlinkX; MeroX; StavroX) that enable the identification of peptides directly from mass spectrometric data, and thus do not rely on gel electrophoresis or immunoblotting [113].

Co-sedimentation analysis of the dystrophin complex in combination with mass spectrometry identified a potential interaction with desmoglein [24]. The dystrophin–glycoprotein complex was enriched by established methods using an advanced subcellular membrane fractionation protocol [15] and detergent solubilization [6], followed by ion exchange and lectin affinity chromatography [7] and a final density gradient ultracentrifugation step [87]. The dystrophin complex was then separated by gradient gel electrophoresis and protein bands digested by on-membrane trypsination [20]. The co-sedimented proteins in the enriched dystrophin fraction were identified by peptide mass spectrometry. Besides the confirmatory identification of dystrophin and its tightly associated glycoproteins by a large number of unique peptide sequences, cytokeratin and the two cytolinker proteins desmoglein and desmoplakin [122] were shown to be present in the dystrophin fraction [24]. This finding agrees with the cell biological concept of the dystrophin complexome being involved in providing a structural lattice for linking individual components of the cytoskeletal network in skeletal muscle fibers. This includes dystrophin interactions with cytokeratin [123,124], synemin [125], plectin [126,127,128], as well as actinin, desmin and tubulin [129]. Intermediate filaments that contain the cytokeratin isoform K19 were shown to directly interact with the actin-binding region of Dp427-M [124].

Dystrophin was established to be directly involved in the organization and stabilization of costameric microtubules [129]. These diverse interactions of dystrophin with intermediate filaments and microtubules suggest that the Dp427-M isoform functions as a cytolinker molecule that maintains close interactions between the sarcolemmal dystroglycan complex and the intracellular cytoskeleton in contractile fibers. In relation to other cytosolic proteins, a close linkage of the dystrophin complex was shown with plakoglobin, which in turn binds to the insulin receptor [130], and the neuronal isoform nNOS of nitric oxide synthase [131]. Both proposed protein interactions provide key hubs for cellular signaling mechanisms within contractile fibers, such as organization of caveolae and the regulation of skeletal muscle size. Dystrophin-associated nitric oxide synthase, the enzyme responsible for the production of the key signaling molecule nitric oxide, was shown to play a regulatory role in costameres [132] and is involved in muscle fatigue, muscular atrophy and certain forms of muscular dystrophy [36,133,134]. In skeletal muscle tissues, nitric oxide levels are low under resting conditions and the production rate of this signaling molecule is greatly enhanced during repetitive muscle contractions. Crucial regulatory functions of nitric oxide in skeletal muscle include auto-regulation of blood flow, metabolic and bioenergetic integration at the level of glucose homeostasis and mitochondrial respiration, modulation of excitation–contraction coupling and contractile force generation, as well as myocyte differentiation [135,136]. The influence of nitric oxide on skeletal muscle function involves changes in redox-sensitive protein species, the activation of the second messenger molecule cyclic guanosine monophosphate and interactions with reactive oxygen species [135].

The localization and continued functioning of nNOS has a crucial regulatory impact on arteriolar blood flow within skeletal muscles. This is clearly shown by the fact that functional ischemia occurs in connection with reduced nitric oxide levels. Narrowing of blood vessels results in oxygen deficiency which in turn renders muscle fibers more susceptible to metabolic stress and cellular degeneration in muscular dystrophy [137,138,139]. Stabilization of nNOS enzyme at the sarcolemma depends on the structural and functional integrity of the sarcoglycan subcomplex. The importance of nNOS-associated signaling mechanisms is highlighted by the pathophysiological consequence of the lack of this enzyme in sarcoglycanopathy, which results in a severely dystrophic phenotype [133,134]. Disturbed allosteric interactions between phosphofructokinase and nNOS were shown to occur in dystrophin-deficient fibers [140], which may contribute to abnormal glycolytic activity patterns in muscular dystrophy [141,142]. The above listed studies allowed to draw the model of the spatial configuration of the dystrophin complex presented in the lower panel of above Figure 1.

A clear linkage to the extracellular matrix via interactions between the dystrophin/dystroglycan-containing sarcolemma and the basal lamina and its connections to layers of different types of collagen is supported by the localization of fibronectin and biglycan in close proximity to dystrophin and utrophin [24,143,144]. Both proteins are key extracellular matrix proteins and are majorly involved in the stabilization of damaged muscle fibers [24,145]. Biglycan appears to be critical for the expression levels of dystrobrevins, syntrophins and sarcoglycans [146,147], which represent dystrophin-associated proteins of high abundance that are routinely identified in crude muscle extracts and purified preparations of the dystrophin–glycoprotein complex [14,24,111,112,148]. In contrast, the highly hydrophobic protein sarcospan of 25 kDa [149] is more difficult to characterize by mass spectrometry. This component of the dystrophin complex is usually only recognized by a small number of unique peptides [19]. Interestingly, studies on subcomplex formation between sarcospan and sarcoglycans indicate a potential indirect linkage via these proteins between the dystrophin complex and the integrin complex of the sarcolemma [150,151].

### 3.2. The Dystrophin Complex as a Cellular Signaling Node in Skeletal Muscle

Besides providing the above-described stabilizing linkage between the basal lamina and the membrane cytoskeleton and thereby functioning as a molecular shock absorber, the dystrophin complex also acts as a critical hub for cellular signaling at the muscle plasma membrane [61]. The dystrophin complexome has been implicated to be involved in the modulation of hypertrophy, major kinase signaling cascades, the organization of caveolae structures, the regulation of skeletal muscle size, the mitogen-activated protein kinase pathway, the regulation of ion homeostasis, cytoskeletal organization, G-protein signaling and neuromuscular transmission in conjunction with its autosomal homologue utrophin, as well as mechano-sensing and cytoskeletal remodeling in association with the laminin-collagen bridge and the sarcolemmal integrin complex [36,61,152].

Following the characterization of dystrophin–dystroglycan interactions [153], it was shown that adaptor proteins of signal transduction and cell communication, such as the growth factor receptor-bound protein Grb2, are also associated with the dystrophin complex [154]. This finding linked dystrophin with cellular signaling processes and was followed by a large number of detailed studies into the physiological and cell biological role of the dystrophin complex. The signaling and regulatory function of the dystrophin complexome in skeletal muscle was shown to involve key physiological players of the muscle plasmalemma, such as channels for potassium, sodium, calcium and water [60]. This includes especially interactions between alpha-syntrophin and inward rectifier K^+^-channels (K_ir_2.1 at the neuromuscular junction) [155], Na^+^-channels (Na_v_1.4 and Na_v_1.5) [93], non-specific channels of the transient receptor potential cation channel family (TRPC1 and TRPC4) [156], and the aquaporin water channel (AQ4) [157,158]. Imaging by confocal microscopy techniques suggests furthermore co-localization of dystrophin and the voltage-sensing dihydropyridine receptor L-type Ca^2+^-channel (Ca_v_1.1) [159]. The dystrophin complex appears to be intrinsically involved in the provision of a structural scaffold for the positioning of important ion channels and transporters.

Kinase-coupled pathways were demonstrated to be coupled to syntrophins and dystroglycans. This includes potential interactions between the non-receptor tyrosine kinase Src and beta-dystroglycan [160,161], as well as binding of various syntrophins to diacylglycerol kinase-zeta [162,163], microtubule-associated serine/threonine kinase MAST205 [164], and a specific mitogen-activated protein kinase named stress-activated protein kinase SAPK3 [165]. The proposed indirect linkage between the dystrophin complex and integrin [166] would suggest that the dystrophin complexome is also partially involved in mechano-sensing and the overall regulation of cytoskeletal remodeling via the phosphatidylinositol 3-kinase-related kinase mTOR (mechanistic/mammalian target of rapamycin) signaling cascade [167]. Other enzymes and regulatory factors involved in dystrophin-linked signaling mechanisms were identified as the ankyrin repeat-rich membrane spanning protein ARMS at the neuromuscular junction [168] and the nNOS isoform of nitric oxide synthase, which is involved in Ca^2+/^calmodulin dependent NO synthesis [36,140]. Myocilin, a modulator of muscle hypertrophy was shown to be linked to alpha-syntrophin [169]. Interestingly, alpha-dystrobrevin was demonstrated to interact with the cytoskeletal linker protein alpha-catulin, which acts as a scaffold structure for G-protein signaling pathways, [170,171,172], and is also linked to liprin and the guanidine nucleotide exchange factor Arhgef5 at the neuromuscular junction [173,174].

Although this article focuses on the dystrophin isoform Dp427-M and its associated components, it is important to briefly mention the crucial function of its autosomal homologue, the dystrophin-related protein utrophin of 395 kDa [175] and the overlapping localization of both membrane cytoskeletal proteins at the neuromuscular junction [176]. In analogy to dystrophin, utrophin also exists in a variety of isoforms, including Up71, G-utrophin, Up140, B-utrophin and A-utrophin. The domain structure of full-length utrophin is very similar to Dp427, especially at the carboxy-terminus, and is characterized by a major actin-binding site at the amino-terminus, a central spectrin-like rod domain and a carboxy-terminal interaction site with beta-dystroglycan [177]. At the motor endplate region, a very high density of utrophin isoform Up395-M exists at the post-synaptic membrane besides dystrophin [176,178]. Utrophin forms a tight complex with proteins involved in the regulation and maintenance of neurotransmission, such as agrin, the muscle-specific kinase MuSK, perlecan, acetylcholinesterase and the nicotinic acetylcholine receptor [179,180]. Figure 5 provides an overview of the complexity of the dystrophin-associated signaling processes and their potential interconnectivities.

## 4. Conclusions

Since the discovery of dystrophin over 30 years ago, a large body of scientific evidence has been gathered that strongly supports the idea that the full-length dystrophin isoform Dp427-M forms the core of a supramolecular protein assembly at the sarcolemma. This dystrophin complexome was shown to be intrinsically involved in linking the extracellular matrix component laminin-211 to the actin membrane cytoskeleton. The main members of the dystrophin-associated complex were identified as sarcolemmal proteins (beta-dystroglycan, alpha-sarcoglycan, beta-sarcoglycan, gamma-sarcoglycan, delta-sarcoglycan and sarcospan), extracellular proteins (alpha-dystroglycan and laminin-211) and cytosolic proteins (alpha-syntrophin, beta-syntrophin, alpha-dystrobrevin, beta-dystrobrevin and cortical actin). Closely interacting proteins with this dystrophin core complex are biglycan and collagens via their coupling to laminin in the extracellular matrix, and various cytoskeletal proteins through the intracellular network of intermediate filaments and microtubules with the actin membrane cytoskeleton. In addition, a variety of enzymes, ion channels and regulatory factors were found to exist in close proximity to dystrophin. These findings favor the systems biological concept of an integrative function of the dystrophin node in skeletal muscle tissues [181]. Dystrophin and its closely associated protein components appear to play a key role in cytoskeletal organization, the maintenance of fiber stability during cellular stresses caused by repeated excitation–contraction–relaxation cycles, the transmission of lateral force throughout the contracting muscle fiber system, and the provision of a structural basis for complex cellular signaling mechanisms.

## Figures and Tables

**Figure 1 proteomes-09-00009-f001:**
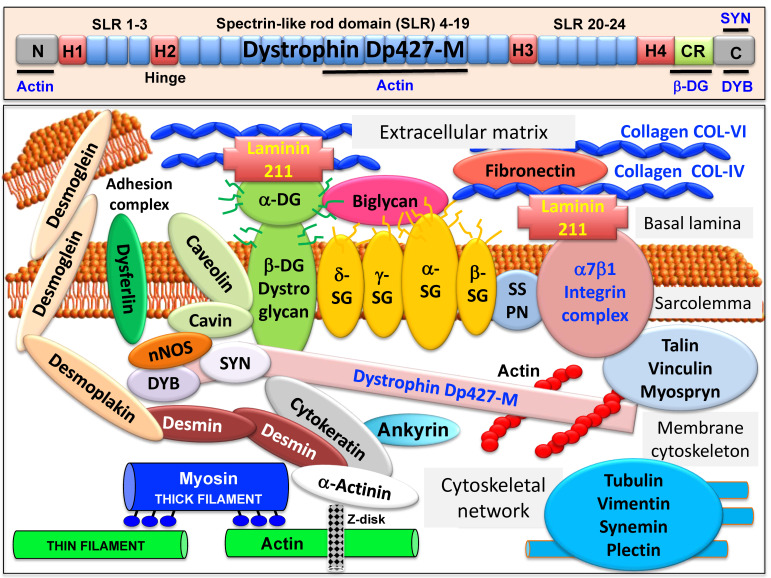
Overview of the domain structure of dystrophin and the diverse interactions of the dystrophin–glycoprotein complex in skeletal muscle tissues. The upper panel shows a diagrammatic presentation of the main molecular domains of dystrophin isoform Dp427-M, including actin-binding sites at the N-terminus and central rod domain, proline-rich hinge regions (H1 to H4), spectrin-like rod (SLR) domains 1–3, 4–19 and 20–24, a cysteine-rich domain with binding sites for integral beta-dystroglycan (DG), the cysteine-rich domain (CR) and the C-terminus with binding sites for dystrobrevin (DYB) and syntrophin (SYN). The lower panel shows a model of the spatial configuration of the dystrophin complexome in skeletal muscle fibers. Shown is the dystrophin core complex consisting of the dystrophin isoform Dp427-M, dystroglycans (DG), sarcoglycans (SG), sarcospan (SSPN), syntrophins (SYN) and dystrobrevins (DYB), as well as the wider dystrophin-associated network that forms associations with the extracellular matrix, the sarcolemma, the cytoskeleton and the sarcomere.

**Figure 2 proteomes-09-00009-f002:**
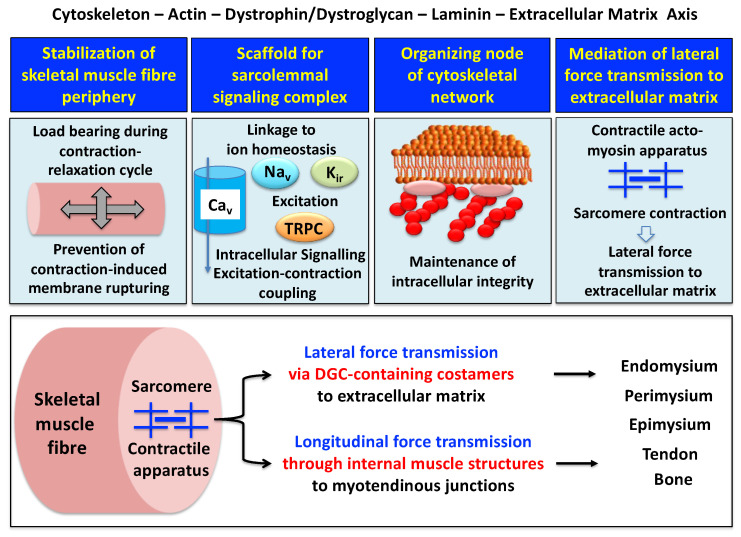
Outline of the main functions of the dystrophin node and its associated protein complex as integrators of fiber stability, cellular signaling. cytoskeletal organization and lateral force transmission. The upper panels summarize the main functions of the trans-sarcolemmal axis formed by the intracellular actin cytoskeleton, the dystrophin–dystroglycan complex, the basal lamina component laminin and the extracellular matrix. The lower panel illustrates the physiological concept of force transmission in skeletal muscles, which can be divided into a laterally and a longitudinally acting system. In conjunction with other costameric proteins, the dystrophin–glycoprotein complex (DGC) is majorly involved in lateral force transmission to the extracellular matrix.

**Figure 3 proteomes-09-00009-f003:**
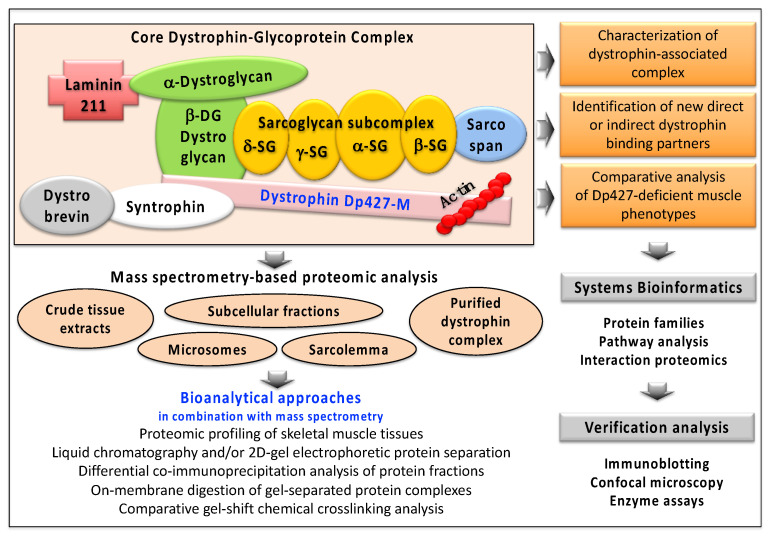
Outline of the biochemical, proteomic and bioinformatic strategy to identify and characterize novel binding proteins of the dystrophin complexome. Mass spectrometry-based proteomic studies have used various starting materials, such as crude tissue extracts, subcellular fractions or purified protein complexes. Listed are the various bioanalytical techniques that are routinely used for the efficient separation of dystrophin-associated protein populations and their subsequent mass spectrometric and bioinformatic evaluation. To verify proteomic findings, often immuno-blotting, biochemical assays and cell biological methods are used.

**Figure 4 proteomes-09-00009-f004:**
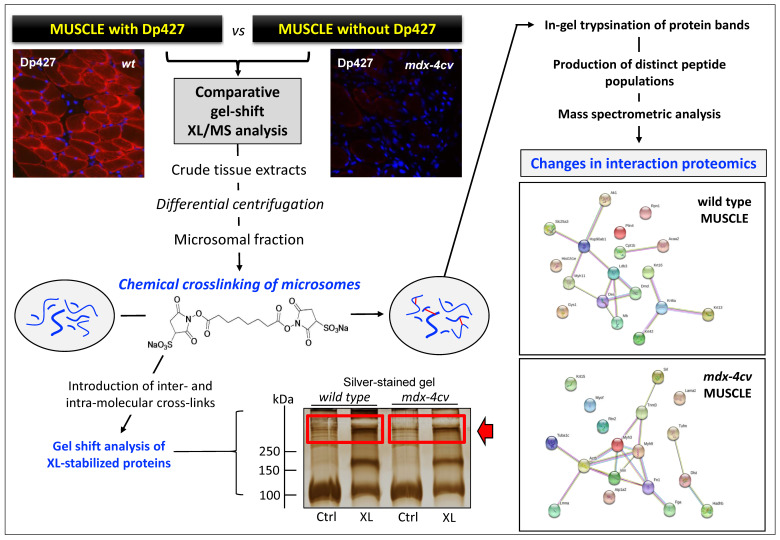
Gel shift-based chemical crosslinking mass spectrometry (XL-MS) analysis of skeletal muscle specimens. Shown is the flow chart of a typical interaction proteomic and bioinformatic analysis using a combination of chemical crosslinking with the 11.4 Å agent BS^3^ (bis [sulfosuccinimidyl] suberate), gel electrophoretic separation of crosslinked protein complexes, mass spectrometric analysis for the identification of proteins with a crosslinker-induced shift in electrophoretic mobility and the bioinformatic evaluation of potential protein–protein interactions. As previously described in detail [112], the various experimental steps involved in this particular type of gel-shift XL-MS analysis include (i) the preparation of crude skeletal muscle tissue extracts by homogenization, (ii) the enrichment of the microsomal membrane fraction using differential centrifugation, (iii) the joining of proteins by incubation with a homo-bifunctional, water-soluble, non-cleavable and amine-reactive crosslinking agent, (iv) the quenching of the crosslinking reaction, (v) extraction of crosslinked proteins and gel electrophoretic separation, (vi) excision of crosslinked protein bands of high molecular mass, (vii) in-gel proteolysis of proteins using trypsin, (viii) separation of peptides by liquid chromatography, (ix) mass spectrometric identification of proteins, and (x) systems bioinformatic analysis of altered protein–protein interaction patterns. The position of the analyzed high-molecular-mass protein bands, using comparative mass spectrometry, are highlighted by a red box and are marked by an arrow. Protein interaction patterns were visualized with the bioinformatic STRING program [121]. The immunofluorescence microscopical images show transverse section of wild-type versus dystrophic *mdx-4cv* mouse *gastrocnemius* muscle. The sarcolemmal localization of dystrophin was stained in red with an antibody to isoform Dp427-M and nuclei were counterstained in blue with the fluorescent stain DAPI (4′,6-diamidino-2-phenylindole) that labels DNA molecules [111].

**Figure 5 proteomes-09-00009-f005:**
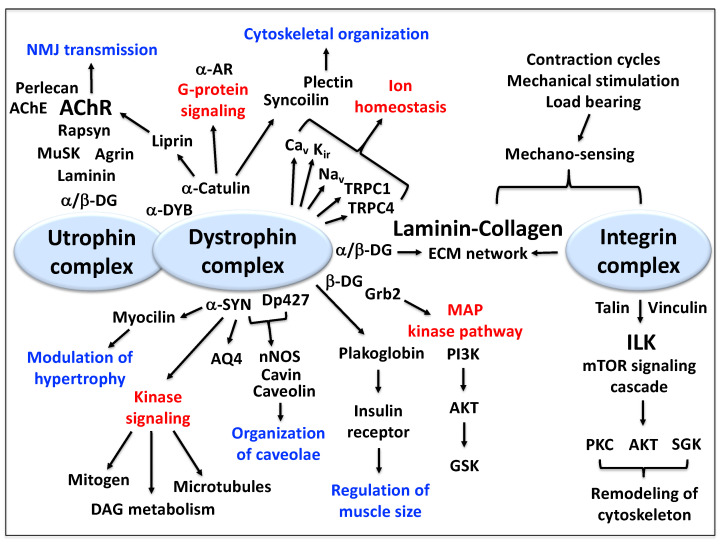
Overview of the function of the dystrophin complexome as a cellular signaling hub at the sarcolemma of skeletal muscle fibers. Shown is the close neighborhood relationship between the dystrophin core complex and the integrin complex within the plasma membrane, as well as the presence of the dystrophin and utrophin complex at the neuromuscular junction. Key pathways and functional results that are potentially linked to dystrophin are marked in red and blue, respectively. Abbreviations used: AChE, acetylcholinesterase; AChR, acetylcholine receptor; AKT, serine/threonine-specific protein kinase B; AQ4, aquaporin water channel; AR, adrenergic receptor; Ca_v_, voltage-sensing L-type Ca^2+^-channel; DAG, diacylglycerol; DG, dystroglycan; Dp427, dystrophin protein of 427 kDa; DYB, dystrobrevin; ECM, extracellular matrix; Grb2, growth factor receptor-bound protein 2; GSK, glycogen synthase kinase; ILK, integrin-linked kinase; K_ir_, inward rectifier K^+^-channel; MAP, mitogen-activated protein; mTOR, mammalian/mechanistic target of rapamycin; MuSK, muscle-specific kinase; Na_v_, Na^+^-channel; NMJ, neuromuscular junction; nNOS, neuronal isoform of nitric oxide synthase; PI3K, phosphoinositide 3-kinase; PKC, protein kinase C; SGK, serine/threonine-protein kinase; SYN, syntrophin; TRPC, transient receptor potential cation channel.

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
