# Peer review of "The Dystrophin Node as Integrator of Cytoskeletal Organization, Lateral Force Transmission, Fiber Stability and Cellular Signaling in Skeletal Muscle"

_proteomes, 2021, doi:10.3390/proteomes9010009_

Round 1
Reviewer 1 Report
This review is original and provides valuable information, especially for the many scientists involved in medical research on Duchenne or Becker muscular dystrophy, often focused on model development, clinical description of the disease, identification of biomarkers, development of therapeutic strategies, and perhaps neglecting the background research conducted to better understand the molecular organizing function of the muscle isoform of dystrophin. There are more than 8000 reviews on this protein and this one is not a simple update of an earlier version. On the contrary, it gathers in an unprecedented way all the data that contributed to the identification of the molecular partners of the complexome centered on dystrophin, with a proposed grouping around four cytoarchitectural and functional functions within the muscle fiber.
The manuscript is well organized, pleasantly written and densely referenced (164 articles cited). The last author of the journal is remarkably well qualified to complete this synthesis. Since his discovery 30 years ago, he has been involved in research on the dystrophin protein and he is an expert in the proteomic analyses performed on muscle fiber.
Nevertheless, I had some difficulties while reading the review and commend to the authors some modifications that will facilitate a more fluid reading of the final document.
Major points
line 137: it would be useful to describe what the lateral transmission force is and its role during the contraction of the muscle fiber. This physical notion which is part of the title is abstruse for a non-specialist.
Figures: this is the weak point of the review. In particular, Figures 1 and 3 are difficult to understand.
Figure 1B is poorly designed. What is called "Skeletal muscle" and resembles a round fiber is in fact the sarcolemma, but then why is the dystrophin not in a sub-sarcolemmal position? This information is partially redundant with Figure 4, and at the same time is not identical. This is awkward. The information in the four blue ellipses is also not in the same order as in the text, and with partially different wordings. We get confused.
Moreover, most of the information in Figure 4 would be useful for understanding the text from the beginning of the review. However, the figure is not mentioned in this part of the text, so you have to think about it yourself and to repeatedly turn pages in order to access this useful information. Some abbreviations are missing for this Figure 4 (e.g. SSPN).
Some fonts used for the figures are not pleasant to read (for example "Spectrin-like rod domain" in Figure 1A), at least on the pdf version I received. Perhaps this is a problem related to the conversion to pdf.
Figure 1 - it may be a good idea to redo Figure 1 by keeping panel 1A and replacing panel 1B with the current Figure 4, which would become 1B and would be cited in the text starting at line 137.
The four groups of functions performed by the dystrophin node would benefit from being arranged from left to right and cited in that order in the text (lines 108-114). The proteins involved in each of these four groups could be of the same color (or with an identical outline), which would make these groups stand out better.Some proteins with dual roles could be grayed out or with a mixed color compatible with the groups (yellow/blue => green).
The Figure could be enlarged to show, for example, Nav1.5/Cav1.1 and illustrate this series of interactions important for excitation-contraction coupling.
Figure 2 - use the same color code for Figure 2
Figure 3 - I did not understand the link between the gel and the right side of the Figure. Are the trypsin complexes those that correspond to the band that appears in the XL versus Ctrl lanes? If so, you have to clarify this by making a frame around the band with an arrow from this frame to the text "In-gel trypsination....". In the current version, the gel seems disconnected from the rest of the protocol, which is not consistent.
Figure 4 (current Figure 5).
The red on blue background reads badly.
It is indicated in the legend that the pathways are in red, but there is a mixture between molecular pathways (G-protein signaling, MAP kinase pathway) and the functional result of activating these pathways (Regulation of muscle size). The two must be prioritized and separated, each with a different color.
Minor points
Alpha, beta, gamma, delta are missing throughout the text... Be careful with the conversion problem when generating the pdf. Change the font if necessary.
line 38: "identification of one of the largest", as DMD is no more the largest gene.
line 57: "primarily affects striated muscles [26], causing respiratory dysfunction, late-onset cardiomyopathy and scoliosis. Functional disturbances etc.".
line 252: "chemical crosslinking mass spectrometry (XL-MS) approach".
line 296: change to "The above listed studies allowed to draw the model of the spatial configuration of the dystrophin complex presented in Figure 1B.".
lines 424, 427, 626: journal name should be italicized.
Good luck for these corrections which I think should be useful.
Author Response
Point-by point response to Reviewer 1:
Comment 1: ‘This review is original and provides valuable information, especially for the many scientists involved in medical research on Duchenne or Becker muscular dystrophy, often focused on model development, clinical description of the disease, identification of biomarkers, development of therapeutic strategies, and perhaps neglecting the background research conducted to better understand the molecular organizing function of the muscle isoform of dystrophin. There are more than 8000 reviews on this protein and this one is not a simple update of an earlier version. On the contrary, it gathers in an unprecedented way all the data that contributed to the identification of the molecular partners of the complexome centered on dystrophin, with a proposed grouping around four cytoarchitectural and functional functions within the muscle fiber. The manuscript is well organized, pleasantly written and densely referenced (164 articles cited). The last author of the journal is remarkably well qualified to complete this synthesis. Since his discovery 30 years ago, he has been involved in research on the dystrophin protein and he is an expert in the proteomic analyses performed on muscle fiber. Nevertheless, I had some difficulties while reading the review and commend to the authors some modifications that will facilitate a more fluid reading of the final document’.
Response: We would like to thank Reviewer 1 for the very positive assessment and helpful suggestions to improve our review. Please find below a point-by-point response to individual comments.
Comment 2: Major point, ‘line 137: it would be useful to describe what the lateral transmission force is and its role during the contraction of the muscle fiber. This physical notion which is part of the title is abstruse for a non-specialist’.
Response: We agree and have added a description of the physiological concept of ‘lateral force transmission’. In order to address the below points on issues with the original figures made by several reviewers, part (B) of Figure 1 was removed and substituted with original Figure 4. The importance of dystrophin in relation to muscle force generation is now better described both in the revised text and also in the lower panel of new Figure 2 in Section 2.2. on the sarcolemmal dystrophin complex and lateral force transmission. See in below response the description of new Fig. 2.
Revised text: ‘At specialized costamere regions, which play both a mechanical and a signaling role, the dystrophin complex forms in conjunction with the integrin-vinculin-talin axis a link to the contractile sarcomere units [63,71]. This bridging structure is postulated to provide an indirect means of lateral force transmission to the collagen-rich muscle exterior, in addition to longitudinal forces that are transmitted directly from the contractile apparatus through the cytosol to the myotendinous junction [72-74]. In skeletal muscle fibres, the characteristic longitudinal pattern of A-bands and I-bands reflect the organization of myosin-containing thick filaments and actin-containing thin filaments with their contractile sarcomeric units, which are positioned between Z-discs. Following the energy-dependent cross-bridge coupling between myosin heads and actin filaments, thin filaments slide past thick filaments. The force generated by this sarcomeric shortening event is partially transmitted in a direct lateral direction between Z-disk structures and the M-line regions of neighboring myofibrils. Costamere structures at the fibre periphery play a central role as sensors of the relative mechanical load and support force transduction across the muscle plasma membrane. Contractile force is then further transmitted to the complex layers of the extracellular matrix, consisting of endomysium, perimysium and epimysium, towards the tendon and bone structure [65,72,73]. The second type of force transmission mechanism works by longitudinal means through internal muscle structures embedded in the cytosol. Both lateral and longitudinal coupling mechanisms act in parallel and ultimately transmit the force generated by the acto-myosin apparatus in the sarcomere to the tendon and anchoring structures such as bone, as diagrammatically summarized in the lower panel of Figure 2. The dystrophin-associated dystroglycan subcomplex was shown to play a critical role in the sarcomeric cytoskeleton by limiting contraction-induced injury to skeletal muscle fibers [70]’.
Comment 3: Major point, ‘Figures: this is the weak point of the review. In particular, Figures 1 and 3 are difficult to understand’.
Response: We agree and have changed the figures accordingly, as outlined in more detail in below responses. Panel 1B was removed from Figure 1 and substituted with the revised image of original Figure 4. New Figure 2 was completely re-designed to address concerns by the reviewers. The colour coding and organization of sarcoglycans was revised in new Fig. 3. Original Figure 3, now new Fig. 4, was revised to highlight the protein bands used for mass spectrometric analysis of crosslinked proteins.
Comment 4: Major point, ‘Figure 1B is poorly designed. What is called "Skeletal muscle" and resembles a round fiber is in fact the sarcolemma, but then why is the dystrophin not in a sub-sarcolemmal position? This information is partially redundant with Figure 4, and at the same time is not identical. This is awkward. The information in the four blue ellipses is also not in the same order as in the text, and with partially different wordings. We get confused’.
Response: We agree and have removed panel 1B from Figure 1 and substituted this part of the figure with the image of the dystrophin complex from original Figure 4. A new Figure 2 with a better organization of the illustration originally presented as panel 1B is now provided in the revised manuscript. As suggested, the information now presented in new Figure 2 explains in a clearer way the multi-functionality of the dystrophin complex, as well as the concept of muscle force transmission.
Comment 5: Major point, ‘Moreover, most of the information in Figure 4 would be useful for understanding the text from the beginning of the review. However, the figure is not mentioned in this part of the text, so you have to think about it yourself and to repeatedly turn pages in order to access this useful information. Some abbreviations are missing for this Figure 4 (e.g. SSPN)’.
Response: To address this issue, missing abbreviations have been added and the lower panel of revised Figure 1 now shows the image of the dystrophin complex from original Figure 4. This should improve the manuscript by introducing already in the first figure the general structure of the wider dystrophin-associated complex in skeletal muscles.
Revised figure legend: ‘Figure 1. Overview of the domain structure of dystrophin and the diverse interactions of the dystrophin-glycoprotein complex in skeletal muscle tissues. The upper panel shows a diagrammatic presentation of the main molecular domains of dystrophin isoform Dp427-M, including actin binding sites at the N-terminus and central rod domain, proline-rich hinge regions (H1 to H4), spectrin-like rod (SLR) domains 1-3, 4-19 and 20-24, a cysteine-rich domain with binding sites for integral beta-dystroglycan (DG), the cysteine-rich domain (CR) and the C-terminus with dystrobrevin (DYB) and syntrophin (SYN) binding sites. The lower panel shows a model of the spatial configuration of the dystrophin complexome in skeletal muscle fibres. Shown is the dystrophin core complex consisting of the dystrophin isoform Dp427-M, dystroglycans (DG), sarcoglycans (SG), sarcospan (SSPN), syntrophins (SYN) and dystrobrevins (DYB), as well as the wider dystrophin-associated network that forms associations with the extracellular matrix, the sarcolemma, the cytoskeleton and the sarcomere.’.
Comment 6: Major point, ‘Some fonts used for the figures are not pleasant to read (for example "Spectrin-like rod domain" in Figure 1A), at least on the pdf version I received. Perhaps this is a problem related to the conversion to pdf’.
Response: There appears to be a general issue with an unintended conversion of symbols and certain characters in the reviewer’s copy of our manuscript. The text and figures were generated on a MacBook-Pro and then uploading to the journal. PDF conversion appears to have introduced unusual alterations in font and symbols in both text and images. In the revised version of the text, we have tried to avoid Macintosh-generated symbols.
Comment 7: Major point, ‘Figure 1 - it may be a good idea to redo Figure 1 by keeping panel 1A and replacing panel 1B with the current Figure 4, which would become 1B and would be cited in the text starting at line 137’.
Response: As already outlined above, we have followed this helpful advice of Reviewer 1 and substituted Fig. 1B with Figure 4.
Comment 8: Major point, ‘The four groups of functions performed by the dystrophin node would benefit from being arranged from left to right and cited in that order in the text (lines 108-114). The proteins involved in each of these four groups could be of the same color (or with an identical outline), which would make these groups stand out better. Some proteins with dual roles could be grayed out or with a mixed color compatible with the groups (yellow/blue => green)’.
Response: In order to address this point, we have re-designed this information on dystrophin functions. New Figure 2 has now adapted the same order of mentioning the dystrophin functions as in the text.
Revised Figure 2: ‘Figure 2. Outline of the main functions of the dystrophin node and its associated protein complex as integrators of fiber stability, cellular signaling. cytoskeletal organization and lateral force transmission. The upper panels summarize the main functions of the trans-sarcolemmal axis formed by the intracellular actin cytoskeleton, the dystrophin/dystroglycan-complex, the basal lamina component laminin and the extracellular matrix. The lower panel illustrates the physiological concept of force transmission in skeletal muscles, which can be divided into a laterally and a longitudinally acting system. In conjunction with other costameric proteins, the dystrophin-glycoprotein complex (DGC) is majorly involved in lateral force transmission to the extracellular matrix.’.
Comment 9: Major point, ‘The Figure could be enlarged to show, for example, Nav1.5/Cav1.1 and illustrate this series of interactions important for excitation-contraction coupling’.
Response: We agree and have added this information in the illustrations shown in new and revised Figure 2, which now contains the link to ion homeostasis and the regulation of excitation-contraction coupling.
Comment 10: Major point, ‘Figure 2 - use the same color code for Figure 2’.
Response: To address this point, we have changed the colouring in original Figure 2, which is now Figure 3 in the revised version of our manuscript, and now uses the same colour code for the different protein components of the dystrophin complex as in the lower panel of Figure 1.
Comment 11: Major point, ‘Figure 3 - I did not understand the link between the gel and the right side of the Figure. Are the trypsin complexes those that correspond to the band that appears in the XL versus Ctrl lanes? If so, you have to clarify this by making a frame around the band with an arrow from this frame to the text "In-gel trypsination....". In the current version, the gel seems disconnected from the rest of the protocol, which is not consistent’.
Response: We agree and have revised this figure, now Figure 4, accordingly. As suggested, the revised figure now contains an arrow that highlights boxes which frame the analysed high-molecular-mass bands following chemical cross-linking and gel electrophoretic separation. The figure legend was changed accordingly:
Revised figure legend: ‘Figure 4. Gel shift based chemical crosslinking … protein-protein interactions. The position of the analysed high-molecular-mass protein bands, using comparative mass spectrometry, are highlighted by a red box and are marked by an arrow. The various experimental steps …’.
Comment 12: ‘Major point, ‘Figure 4 (current Figure 5). The red on blue background reads badly’.
Response: We agree and have changed accordingly the colour and writing in original Figure 5 in the revised version of our manuscript.
Comment 13: Major point: ‘It is indicated in the legend that the pathways are in red, but there is a mixture between molecular pathways (G-protein signaling, MAP kinase pathway) and the functional result of activating these pathways (Regulation of muscle size). The two must be prioritized and separated, each with a different color’.
Response: The colour coding has been changed in revised Figure 5.
Revised figure legend: ‘Figure 5. Overview of the function of the dystrophin complexome as a cellular signaling hub at the sarcolemma of skeletal muscle fibers. … Key pathways and functional results that are potentially linked to dystrophin are marked in red and blue, respectively. Abbreviations …’.
Comment 14: Minor point. ‘Alpha, beta, gamma, delta are missing throughout the text... Be careful with the conversion problem when generating the pdf. Change the font if necessary’.
Response: There appears to be a general issue with an unintended conversion of symbols and in the reviewer’s copy of our manuscript. In the revised version of the text, we have tried to avoid Macintosh-generated symbols.
Comment 15: Minor point, ‘line 38: "identification of one of the largest", as DMD is no more the largest gene’.
Response: This point has ben addressed. The first sentence in the Introduction section now reads as follows: ‘Following the identification of one of the largest genes in the human genome, the DMD gene [1] and the initial characterization of its full-length protein product [2,3], it became clear that dystrophin exists in a variety of isoforms ranging in molecular mass from approximately 71 kDa to 427 kDa [4]’.
Comment 16: Minor point, ‘line 57: "primarily affects striated muscles [26], causing respiratory dysfunction, late-onset cardiomyopathy and scoliosis. Functional disturbances etc".’.
Response: We agree and have changed this part of the text as follows: ‘… The almost complete loss of dystrophin isoform Dp427-M and concomitant reduction of dystrophin-associated proteins trigger the complex pathophysiology of Duchenne muscular dystrophy [7,25], an X-linked and multifaceted disorder of early childhood that primarily affects striated muscles [26], causing respiratory dysfunction, late-onset cardiomyopathy and scoliosis. Functional disturbances of the nervous system [27] and abnormal energy metabolism are also distinctive features of dystrophinopathy [28,29] …’.
Comment 17: Minor point, ‘line 252: "chemical crosslinking mass spectrometry (XL-MS) approach".’.
Response: We agree and have changed this part of the text as follows: ‘… Figure 3 outlines how an chemical crosslinking mass spectrometry (XL-MS) approach can be employed to study novel protein-protein interaction patterns, which are indirectly linked to dystrophin expression levels [112] …’.
Comment 18: Minor point, ‘line 296: change to "The above listed studies allowed to draw the model of the spatial configuration of the dystrophin complex presented in Figure 1B".’.
Response: We agree and have changed this part of the text as follows: ‘ … The above listed studies allowed to draw the model of the spatial configuration of the dystrophin complex presented in the lower panel of Figure 1 …’.
Comment 19: Minor point, ‘lines 424, 427, 626: journal name should be italicized’.
Response: Journal names have been revised and are now shown in italics in these references.
Comment 20: Minor point: ‘Good luck for these corrections which I think should be useful’.
Response: We would like to thank Reviewer 1 for the thorough review of our manuscript and the many constructive suggestions for improving both text and figures.
Reviewer 2 Report
In this review entitled "The dystrophin node as integrator of cytoskeletal organization, lateral force transmission, fiber stability and cellular signaling" the authors describe the central role of full-length muscle isoform of dystrophin (Dp-427-M) and the dystrophin-glycoprotein complex in the cytoskeletal organization, sarcolemmal stabilization and cellular signaling in skeletal muscle. It focuses on the importance of combining different techniques (such as, mass spectrometry-based proteomic, chromatography, and chemical crosslinking) to individuate novel members of the dystrophin complex and study their functions. The manuscript is well written and the topic is interesting. The authors should address the following major and minor points.
Major points
- Figure quality should be improved, except for Figure 3.
- Line 228: Please briefly describe the chemical crosslinking technique.
- Figure 3 legend: the authors should better describe the figure and briefly describe the steps involved in chemical crosslinking mass spectrometry.
- The authors should add a brief description of the Utrophin structure.
Minor points
- Lines 18 and 38 "the Dmd gene": The human DMD gene should be written in uppercase and italics form
- Lines 20 and 42 "Dp427-M isoform": Please specify "muscle isoform”.
- Lines 56-60: The authors state, "the complex pathophysiology of Duchenne muscular dystrophy [7,25], an X-linked and multifaceted disorder of early childhood that primarily affects the skeletal musculature [26]. However, respiratory dysfunction and late-onset cardiomyopathy, as well as scoliosis, functional disturbances of the nervous system and abnormal energy metabolism are also distinctive features of dystrophinopathy [27,28]." This sentence should be rephrased because the respiratory dysfunction and scoliosis are directly linked to skeletal muscle impairment. DMD patients experience progressive muscle degeneration with the loss of ambulation, and die prematurely for respiratory failure and cardiomyopathy.
-Lines 119-120: In the legend of Figure 1A the authors reported "spectrin-like rod (SLR) domains 1-2" instead of “1-3” as described in the figure.
Author Response
Point-by point response to Reviewer 2:
Comment 1: ‘In this review entitled "The dystrophin node as integrator of cytoskeletal organization, lateral force transmission, fiber stability and cellular signaling" the authors describe the central role of full-length muscle isoform of dystrophin (Dp-427-M) and the dystrophin-glycoprotein complex in the cytoskeletal organization, sarcolemmal stabilization and cellular signaling in skeletal muscle. It focuses on the importance of combining different techniques (such as, mass spectrometry-based proteomic, chromatography, and chemical crosslinking) to individuate novel members of the dystrophin complex and study their functions. The manuscript is well written and the topic is interesting. The authors should address the following major and minor points’.
Response: We would like to thank Reviewer 2 for the critical and constructive assessment of our review. Please find below a point-by-point response to individual comments.
Comment 2, Major point: ‘Figure quality should be improved, except for Figure 3’.
Response: We agree and since other reviewers have also outlined issues with the originally submitted figures, the following changes were introduced:
New Fig. 1: Panel 1B was removed from Figure 1 and substituted with the image of original Figure 4 to show already at the beginning of the manuscript the general structure of the wider dystrophin complex.
New Fig. 2: This newly positioned and revised figure explains now in a clearer way the multi-functionality of the dystrophin complex, as well as the concept of muscle force transmission.
New Fig. 3 (originally Fig. 2): The colour coding of individual proteins was changed to agree with the same colouring used in the lower panel of revised Fig. 1 which outlines the configuration of the dystrophin complex. In addition, the configuration of the sarcoglycan sub-complex was revised.
New Fig. 4 (originally Fig. 3): The figure was slightly revised to highlight the protein bands used for mass spectrometric analysis of crosslinked proteins.
New Fig. 5: Colour coding was changed to better differentiate between signaling pathways and functional consequences.
Comment 3, Major point: ‘Line 228: Please briefly describe the chemical crosslinking technique’.
Response: This point has been addressed and the revised manuscript now contains a general description of the chemical crosslinking technique, as follows ‘ … extensive biochemical isolation procedures. In protein biochemistry, chemical crosslinking is defined as the intra- and/or inter-molecular stabilization of protein molecules via covalent bonding. Since the chemical crosslinking technique is capable of determining molecular changes in macromolecular oligomerization, it is an ideal bioanalytical tool for studying dynamic protein interaction patterns. The select joining of two or more protein species is achieved by the incubation of protein mixtures with a large variety of crosslinking agents that differ in their molecular spacer arm length, their solubility and chemical reactivity. Frequently used crosslinkers contain reactive groups that are suitable for optimum interactions with functional groups in protein molecules such as primary amines. For example, the crosslinker used in below Figure 4 is a homo-bifunctional, water-soluble, non-cleavable and amine-reactive agent that is highly useful for joining proteins in complex assemblies under physiological conditions. Following the addition of crosslinking agents, protein samples are usually incubated for 30 minutes at 25oC for 30 minutes and then the reaction quenched by the addition of ammonium acetate. A convenient way to analyze the occurrence of crosslinked protein complexes is the determination of an altered electrophoretic mobility in one- or two-dimensional gels. A crucial issue with this method is the potential occurrence …’.
Comment 4, Major point: ‘Figure 3 legend: the authors should better describe the figure and briefly describe the steps involved in chemical crosslinking mass spectrometry’.
Response: We agree and have added this information in the figure legend of new Figure 4 (previously Fig. 3) as follows: … of potential protein-protein interactions. As previously described in detail [107], the various experimental steps involved in this particular type of gel-shift XL-MS analysis include (i) the preparation of crude skeletal muscle tissue extracts by homogenization, (ii) the enrichment of the microsomal membrane fraction using differential centrifugation, (iii) the joining of proteins by incubation with a homo-bifunctional, water-soluble, non-cleavable and amine-reactive crosslinking agent, (iv) the quenching of the crosslinking reaction, (v) extraction of crosslinked proteins and gel electrophoretic separation, (vi) excision of crosslinked protein bands of high molecular mass, (vii) in-gel proteolysis of proteins using trypsin, (viii) separation of peptides by liquid chromatography, (ix) mass spectrometric identification of proteins, and (x) systems bioinformatic analysis of altered protein-protein interaction patterns. The position of the analysed high-molecular-mass protein bands, using comparative mass spectrometry, are highlighted by a red box and are marked by an arrow. Protein interaction patterns were visualized with …’.
Comment 5, Major point: ‘The authors should add a brief description of the Utrophin structure’.
Response: We agree and have added the following statement to the revised text: ‘ … at the neuromuscular junction [176]. In analogy to dystrophin, utrophin also exists in a variety of isoforms, including Up71, G-utrophin, Up140, B-utrophin and A-utrophin. The domain structure of full-length utrophin is very similar to Dp427, especially at the carboxy-terminus, and is characterized by a major actin-binding site at the amino-terminus, a central spectrin-like rod domain and a carboxy-terminal interaction site with beta-dystroglycan [177]. At the motor endplate region …’.
New Reference [177] Blake, D.J.; Tinsley, J.M.; Davies, K.E. Utrophin: a structural and functional comparison to dystrophin. Brain Pathol. 1996, 6, 37-47.
Comment 6, Minor point: ‘Lines 18 and 38 "the Dmd gene": The human DMD gene should be written in uppercase and italics form’.
Response: The ‘DMD’ gene symbol has been corrected in the text.
Comment 7, Minor point: ‘Lines 20 and 42 "Dp427-M isoform": Please specify "muscle isoform”.’.
Response: To address this point, the text in the Abstract section has been changed to: ‘ … its full-length muscle isoform Dp427-M is …’.
Comment 8, Minor point: ‘Lines 56-60: The authors state, "the complex pathophysiology of Duchenne muscular dystrophy [7,25], an X-linked and multifaceted disorder of early childhood that primarily affects the skeletal musculature [26]. However, respiratory dysfunction and late-onset cardiomyopathy, as well as scoliosis, functional disturbances of the nervous system and abnormal energy metabolism are also distinctive features of dystrophinopathy [27,28]." This sentence should be rephrased because the respiratory dysfunction and scoliosis are directly linked to skeletal muscle impairment. DMD patients experience progressive muscle degeneration with the loss of ambulation, and die prematurely for respiratory failure and cardiomyopathy’.
Response: We agree and have as changed this part of the text as follows: ‘… The almost complete loss of dystrophin isoform Dp427-M and concomitant reduction of dystrophin-associated proteins triggers the complex pathophysiology of Duchenne muscular dystrophy [7,25], an X-linked and multifaceted disorder of early childhood that primarily affects striated muscles [26], causing respiratory dysfunction, late-onset cardiomyopathy and scoliosis. Functional disturbances of the nervous system [27] and abnormal energy metabolism are also distinctive features of dystrophinopathy [28,29] …’.
New reference [27]: Doorenweerd, N. Combining genetics, neuropsychology and neuroimaging to improve understanding of brain involvement in Duchenne muscular dystrophy - a narrative review. Neuromuscul. Disord. 2020, 30, 437-442.
Comment 9, Minor point: ‘Lines 119-120: In the legend of Figure 1A the authors reported "spectrin-like rod (SLR) domains 1-2" instead of “1-3” as described in the figure’.
Response: We would like to thank Reviewer 2 for pointing out this error in the text. This mistake has now been corrected.
Reviewer 3 Report
This review on dystrophin complex describes and covers mostly aspects of its interaction with cytoskeletal and extramuscular matrix proteins in the normal state,there is no tentative or interpretation of interaction of dystrophin isoforms that are mentioned in lines 167-170 and their action in other tissues beside skeletal muscle i.e.CNS and as stated Authors have avoided covering any pathology specific issue,nonetherless this would been an interesting field to explore.
I suggest to integrate this interesting review with the role of dystrophin in CNS as well overlapping features of other myopathies,such as the change of sarcolemma in DMD and the possible association in caveolinopathies and in sarcoglycanopathies,especially in relation to dystrophin complex.
It would be of interest some cosideration on the type of complex that could be assembled and possible protein interactions for lack of isoforms in CNS that determines important structural changes (see Angelini and Pinzan TAND 2019),here there are some observation on presence of dystrophin complex in Purkinje cells etc.is this the same complex that is decribed in muscle?
There is little mention of dystrophin interaction with Nitric Oxide Synthetase or its possible function.The role of NOS seems crucial in exercise and fatigue (Fanin et al.2009) in LGMD but similar observations apply to other dystrophin related myopathies such as BMD.In DMD and sarcoglycanopathies lack of NOS might be resposible for heart involvement,what could be discussed in the involvement of this protein is fatigue and vasodilatation.(see Angelini and Tasca 2015).
Another underepresented field is the interaction waiting further analytic studies with the sarcoglycan complex and its organisation.
I find contradictory the two schematic figures 2 and 4,fig.2 with an alphabetic order of the sarcoglycans is unreal but fig.4 proposed organisarion of sarcoglycan proteins seems in my view more correct in sarcoglycan complex organisation since beta and delta sarcoglycan are the core of the complex( see Angelini et al.2020 ).
There is a need to connect the introduction with some final conclusion on interaction of dystrophin with ion channels and signalling systems.
The connection of dystrophin with caveolin is of interest,since from the preliminary studies of Schotland et al.(1981) with freeze fracture experiments a lack of caveolae was observed in DMD muscle.Another features that deserve attention is the role of dystrophin complex in featuring muscle mass and its relations here proposed with glycogen/glycolitic system ,in biochemical experiments (Dimauro,Catani et al.JNeurol,Neurosurg,Psychiat,1967)this pathway was found particularly defective in biopsies from muscular dystrophy patients.
Minor
in my manuscript version some greek letters are missing I presume that on line 188 the precursor name is alpha/beta dystroglycan.
Author Response
Point-by point response to Reviewer 3:
Comment 1, Major point: ‘This review on dystrophin complex describes and covers mostly aspects of its interaction with cytoskeletal and extramuscular matrix proteins in the normal state, there is no tentative or interpretation of interaction of dystrophin isoforms that are mentioned in lines 167-170 and their action in other tissues beside skeletal muscle i.e. CNS and as stated Authors have avoided covering any pathology specific issue, nonetherless this would been an interesting field to explore’.
Response: We would like to thank Reviewer 3 for the critical assessment of our article and additional suggestions. We agree that many aspects of dystrophin in relation to its diverse function in a variety of tissues and organs other than skeletal muscles, as well as involvement in several pathologies, would also be of interest to many readers. However, our manuscript is an invited contribution with a specific focus on normal dystrophin function in skeletal muscles and the topic has been outlined and provided to the editor prior to submission. The approach of discussing the multi-functional role of the dystrophin complex specifically in normal skeletal muscle, and not mayorly focusing on neither the molecular pathogenesis of Duchenne muscular dystrophy nor dystrophin complexes in other tissues such as the heart or the central nervous system, was agreed. As can be seen from the responses by the other reviewers, especially Reviewer 1 and Reviewer 4, there is agreement on the suitability of this approach with a focus of dystrophin functioning in normal skeletal muscle. We never-the-less agree with Reviewer 3 on the importance of related aspects of dystrophin in other tissues and its pathophysiological role. Accordingly we have addressed suggestions and additional references by Reviewer 3 as outlined in detail in below responses.
Comment 2, Major point: ‘I suggest to integrate this interesting review with the role of dystrophin in CNS as well overlapping features of other myopathies, such as the change of sarcolemma in DMD and the possible association in caveolinopathies and in sarcoglycanopathies, especially in relation to dystrophin complex’.
Response: The complexity of dystrophin involvement in these diseases is now mentioned in revised text, as follows: ‘… In addition to the involvement of sarcolemmal abnormalities in highly progressive Duchenne muscular dystrophy and more benign forms of Becker muscular dystrophy, changes in the dystrophin complex are also involved in sarcoglycanopathies and alpha-dystroglycanopathies. Sarcoglycanopathies represent subtypes of limb-girdle muscular dystrophy that are caused by primary abnormalities in the dystrophin-associated sarcoglycans and alpha-dystroglycanopathies are associated with the abnormal glycosylation of alpha-dystroglycan [38]. Caveolinopathies are another group of disorders that effect the plasma membrane via alterations in caveolae invaginations. Primary abnormalities in caveolin-3 are associated with certain forms of muscular dystrophy and the altered expression of this protein is postulated to also play a pathophysiological role in dystrophinopathy [39]. Importantly, new findings on …’.
New References:
[38] Liewluck, T.; Milone, M. Untangling the complexity of limb-girdle muscular dystrophies. Muscle Nerve. 2018, 58, 167-177.
[39] Pradhan, B.S.; PrószyÅ„ski, T.J. A Role for Caveolin-3 in the Pathogenesis of Muscular Dystrophies. Int. J. Mol. Sci. 2020, 21, 8736.
Comment 3, Major point: ‘It would be of interest some cosideration on the type of complex that could be assembled and possible protein interactions for lack of isoforms in CNS that determines important structural changes (see Angelini and Pinzan TAND 2019),here there are some observation on presence of dystrophin complex in Purkinje cells etc.is this the same complex that is decribed in muscle?’.
Response: To address this comment, we have added relevant papers to the revised manuscript. This includes a new and recent review of brain involvement in dystrophinopathy in the Introduction section as new Reference [27] (Doorenweerd, N. Combining genetics, neuropsychology and neuroimaging to improve understanding of brain involvement in Duchenne muscular dystrophy - a narrative review. Neuromuscul. Disord. 2020, 30, 437-442). We now also quote the above suggested article by Angelini and Pinzan [Angelini C, Pinzan E. Advances in imaging of brain abnormalities in neuromuscular disease. Ther Adv Neurol Disord. 2019;12:1756286419845567], as well as a review on the role of the dystrophin-glycoprotein complex in the central nervous system [Waite, A., Brown, S.C.; Blake, D.J. The dystrophin-glycoprotein complex in brain development and disease. Trends Neurosci. 2012, 35, 487-496.] and a paper on the importance of the complex formation between Dp71 dystrophin, beta-dystroglycan and alpha-syntrophin [82. Naidoo, M.; Anthony, K. Dystrophin Dp71 and the Neuropathophysiology of Duchenne Muscular Dystrophy. Mol. Neurobiol. 2020, 57, 1748-1767] in the revised text and briefly outline the importance of dystrophins in the central nervous system, as follows:
‘… Functional disturbances of the nervous system [27] and abnormal energy metabolism are also distinctive features of dystrophinopathy [28,29] …’.
‘ … central nervous system [84]. Of note, the central nervous system displays one of the greatest varieties of dystrophin isoforms, which are involved in synaptic modulation, neuronal excitability and signal integration. Brain Dp427-B is present in neurons of the cerebral cortex and in cerebellar Purkinje cells, Dp140-B is highly expressed during brain development and Dp71-G is located in both neurons and glia cells in the dentate gyrus [82]. Cognitive impairments and emotional disturbances in Duchenne patients are probably linked to altered dystrophin expression in the central nervous system and this is reflected by structural brain abnormalities [85]. The formation of dystrophin complexes and their involvement in dystrophinopathy-associated brain defects has been reviewed by Waite et al. [86] …’.
New References :
[27] Doorenweerd, N. Combining genetics, neuropsychology and neuroimaging to improve understanding of brain involvement in Duchenne muscular dystrophy - a narrative review. Neuromuscul. Disord. 2020, 30, 437-442.
[82] Naidoo M, Anthony K. Dystrophin Dp71 and the Neuropathophysiology of Duchenne Muscular Dystrophy. Mol Neurobiol. 2020;57(3):1748-1767.
[85] Angelini C, Pinzan E. Advances in imaging of brain abnormalities in neuromuscular disease. Ther Adv Neurol Disord. 2019;12:1756286419845567.
[86] Waite A, Brown SC, Blake DJ. The dystrophin-glycoprotein complex in brain development and disease. Trends Neurosci. 2012;35(8):487-496.
Comment 4, Major point: ‘There is little mention of dystrophin interaction with Nitric Oxide Synthetase or its possible function.The role of NOS seems crucial in exercise and fatigue (Fanin et al.2009) in LGMD but similar observations apply to other dystrophin related myopathies such as BMD.In DMD and sarcoglycanopathies lack of NOS might be resposible for heart involvement,what could be discussed in the involvement of this protein is fatigue and vasodilatation.(see Angelini and Tasca 2015).
Response: We would like to thank Reviewer 3 for highlighting the importance of nNOS. We have integrated the mentioned manuscripts and other relevant papers on nNOS action in skeletal muscle. The revised text contains now a section on nNOS, as follows:
Revised text: ‘ … Dystrophin-associated nitric oxide synthase, the enzyme responsible for the production of the key signaling molecule nitric oxide, was shown to play a regulatory role in costameres [132] and is involved in muscle fatigue, muscular atrophy and certain forms of muscular dystrophy [ 36,133,134]. In skeletal muscle tissues, nitric oxide levels are low under resting conditions and the production rate of this signaling molecule is greatly enhanced during repetitive muscle contractions. Crucial regulatory functions of nitric oxide in skeletal muscle include auto-regulation of blood flow, metabolic and bioenergetic integration at the level of glucose homeostasis and mitochondrial respiration, modulation of excitation-contraction coupling and contractile force generation, as well as myocyte differentiation [135,136]. The influence of nitric oxide on skeletal muscle function involves changes in redox-sensitive protein species, the activation of the second messenger molecule cyclic guanosine monophosphate and interactions with reactive oxygen species [135]. … The localization and continued functioning of nNOS has a crucial regulatory impact on arteriolar blood flow within skeletal muscles. This is clearly shown by the fact that functional ischemia occurs in connection with reduced nitric oxide levels. Narrowing of blood vessels results in oxygen deficiency which in turn renders muscle fibres more susceptible to metabolic stress and cellular degeneration in muscular dystrophy [137-139]. Stabilization of nNOS enzyme at the sarcolemma depends on the structural and functional integrity of the sarcoglycan sub-complex. The importance of nNOS-associated signaling mechanisms is highlighted by the pathophysiological consequence of the lack of this enzyme in sarcoglycanopathy, which results in a severely dystrophic phenotype [133,134]. Disturbed allosteric interactions between phosphofructokinase and nNOS were shown to occur in dystrophin-deficient fibres [140], which may contribute to abnormal glycolytic activity patterns in muscular dystrophy [141,142]. The above listed studies allowed to draw the model of the spatial configuration of the dystrophin complex presented in the lower panel of Figure 1 …‘.
[132] Gorza, L.; Sorge, M.; Seclì, L.; Brancaccio, M. Master Regulators of Muscle Atrophy: Role of Costamere Components. Cells 2021, 10, E61.
[133] Fanin, M.; Tasca, E.; Nascimbeni, A.C.; Angelini, C. Sarcolemmal neuronal nitric oxide synthase defect in limb-girdle muscular dystrophy: an adverse modulating factor in the disease course? J. Neuropathol. Exp. Neurol. 2009, 68, 383-390.
[134] Angelini, C.; Tasca, E.; Nascimbeni, A.C.; Fanin, M. Muscle fatigue, nNOS and muscle fiber atrophy in limb girdle muscular dystrophy. Acta Myol. 2014, 33, 119-126.
[135] Stamler, J.S.; Meissner, G. Physiology of nitric oxide in skeletal muscle. Physiol. Rev. 2001, 81, 209-237.
[136] Gantner, B.N.; LaFond, K.M.; Bonini, M.G. Nitric oxide in cellular adaptation and disease. Redox Biol. 2020, 34, 101550.
[137] Sato, K.; Yokota, T.; Ichioka, S.; Shibata, M.; Takeda, S. Vasodilation of intramuscular arterioles under shear stress in dystrophin-deficient skeletal muscle is impaired through decreased nNOS expression. Acta Myol. 2008, 27, 30-36.
[138] Tidball, J.G.; Wehling-Henricks, M. Nitric oxide synthase deficiency and the pathophysiology of muscular dystrophy. J. Physiol. 2014, 592, 4627-4638.
[139] Nichols, B.; Takeda, S.; Yokota, T. Nonmechanical Roles of Dystrophin and Associated Proteins in Exercise, Neuromuscular Junctions, and Brains. Brain Sci. 2015, 5, 275-298.
[140] Wehling-Henricks, M.; Oltmann, M.; Rinaldi, C.; Myung, K.H.; Tidball, J.G. Loss of positive allosteric interactions between neuronal nitric oxide synthase and phosphofructokinase contributes to defects in glycolysis and increased fatigability in muscular dystrophy. Hum. Mol. Genet. 2009, 18, 3439-3451.
[141] Di Mauro, S.; Angelini, C.; Catani, C. Enzymes of the glycogen cycle and glycolysis in various human neuromuscular disorders. J. Neurol. Neurosurg. Psychiatry 1967, 30, 411-415.
[142] Chinet, A.E.; Even, P.C.; Decrouy, A. Dystrophin-dependent efficiency of metabolic pathways in mouse skeletal muscles. Experientia 1994, 50, 602-605.
Comment 5, Major point: ‘Another underepresented field is the interaction waiting further analytic studies with the sarcoglycan complex and its organisation.’
Response: We agree and have added information on the sarcoglycan complex configuration as follows, quoting the reference [97]: Tarakci, H.; Berger, J. The sarcoglycan complex in skeletal muscle. Front. Biosci. 2016; 21, 744-756.
Revised text: ‘… As reviewed by Tarakci and Berger [97], the sarcoglycan sub-complex is initially assembled by the formation of a core between beta-sarcoglycan and delta-sarcoglycan, which subsequently recruits the other two sarcoglycans. Through interactions with sarcospan and additional dystrophin-associated proteins, the sarcoglycan complex secures the formation and mechanical maintenance of the sarcolemmal dystrophin complex. Besides its integrating role in membrane stabilization, the sarcoglycan sub-complex can be chemically modified during fibre contraction, which provides the transduction of information on relative contractile force into cellular signaling [97] …’.
Comment 6, Major point: ‘I find contradictory the two schematic figures 2 and 4,fig.2 with an alphabetic order of the sarcoglycans is unreal but fig.4 proposed organisarion of sarcoglycan proteins seems in my view more correct in sarcoglycan complex organisation since beta and delta sarcoglycan are the core of the complex( see Angelini et al.2020 ).
Response: To address this point, the sarcoglycan order in original Figure 2, now numbered as new Figure 3, was changed accordingly.
Comment 7, Major point: ‘There is a need to connect the introduction with some final conclusion on interaction of dystrophin with ion channels and signalling systems’.
Response: To address this point, the final sentence of the Introduction section has be revised as follows: ‘… In addition, the dystrophin complex has been implicated to provide a master node for cytoskeletal organization and cellular signaling events, which are characterized by the linkage of dystrophin to ion channels, the insulin signaling pathway, nitric oxide-based regulatory processes, kinase signaling pathways and excitation-contraction coupling’.
Comment 8, Major point: ‘The connection of dystrophin with caveolin is of interest,since from the preliminary studies of Schotland et al.(1981) with freeze fracture experiments a lack of caveolae was observed in DMD muscle. Another features that deserve attention is the role of dystrophin complex in featuring muscle mass and its relations here proposed with glycogen/glycolitic system, in biochemical experiments (Dimauro,Catani et al.JNeurol,Neurosurg,Psychiat,1967) this pathway was found particularly defective in biopsies from muscular dystrophy patients’.
Response: To address these points, the mentioned references are now quoted in different sections of the revised text that are concerned with dystrophin, cavins and caveolae, as well as nNOS signaling and glycolytic enzymes in relation to the dystrophin complex. The paper on caveolae is also quoted in connection with the more recent review on the role of caveolin in the pathogenesis of DMD by Pradhan and PrószyÅ„ski [Int J Mol Sci. 2020, 21, 8736] and the paper on glycolysis is quoted in conjunction with two other relevant papers [Wehling-Henricks, M.; Oltmann, M.; Rinaldi, C.; Myung, K.H.; Tidball, J.G. Loss of positive allosteric interactions between neuronal nitric oxide synthase and phosphofructokinase contributes to defects in glycolysis and increased fatigability in muscular dystrophy. Hum. Mol. Genet. 2009, 18, 3439-3451, Chinet, A.E.; Even, P.C.; Decrouy, A. Dystrophin-dependent efficiency of metabolic pathways in mouse skeletal muscles. Experientia 1994, 50, 602-605].
Revised text: ‘… cellular signaling and mechano-protection [116]. Freeze-fracture electron microscopy has established that the number and structure of caveolae are changed in dystrophinopathy suggesting an important role of abnormal cavins in muscular dystrophy [39, 117]. Of note, the crucial repair protein … ’.
New References:
[39] Pradhan, B.S.; PrószyÅ„ski, T.J. A Role for Caveolin-3 in the Pathogenesis of Muscular Dystrophies. Int. J. Mol. Sci. 2020, 21, 8736.
[117] Bonilla, E.; Fischbeck, K.; Schotland, D.L. Freeze-fracture studies of muscle caveolae in human muscular dystrophy. Am. J. Pathol. 1981, 104,167-173.
Revised text: ‘… in a severely dystrophic phenotype [133,134]. Disturbed allosteric interactions between phosphofructokinase and nNOS were shown to occur in dystrophin-deficient fibres [140], which may contribute to abnormal glycolytic activity patterns in muscular dystrophy [141,142]. The above …’.
New References:
[140] Wehling-Henricks, M.; Oltmann, M.; Rinaldi, C.; Myung, K.H.; Tidball, J.G. Loss of positive allosteric interactions between neuronal nitric oxide synthase and phosphofructokinase contributes to defects in glycolysis and increased fatigability in muscular dystrophy. Hum. Mol. Genet. 2009, 18, 3439-3451.
[141] Di Mauro, S.; Angelini, C.; Catani, C. Enzymes of the glycogen cycle and glycolysis in various human neuromuscular disorders. J. Neurol. Neurosurg. Psychiatry 1967, 30, 411-415.
[142] Chinet, A.E.; Even, P.C.; Decrouy, A. Dystrophin-dependent efficiency of metabolic pathways in mouse skeletal muscles. Experientia 1994, 50, 602-605.Comment 9, Minor point: ‘In my manuscript version some greek letters are missing I presume that on line 188 the precursor name is alpha/beta dystroglycan’.
Comment 9, Minor point: ‘In my manuscript version some greek letters are missing I presume that on line 188 the precursor name is alpha/beta dystroglycan’.
Response: There appears to be a general issue with an unintended conversion of symbols and certain characters in the reviewer’s copy of our manuscript. The text and figures were generated on a MacBook-Pro and then uploading to the journal. PDF conversion appears to have introduced unusual alterations in font and symbols in both text and images. In the revised version of the text, we have tried to avoid Macintosh-generated symbols.
Reviewer 4 Report
The authors present a timely, well structured, and comprehensive review on dystrophin and its role in skeletal muscle.
Suggested edits:
- Change the title to reflect that the review primarily concerns the role of dystrophin in skeletal muscle; while mention is made of other tissues and cardiac muscle, this is not covered in detail.
- Use the correct notation for all genes- in the abstract the notation Dmd is appropriate for mouse, and since DMD patients are being referred to, DMD should be used.
- See also page 9, 4th paragraph. Ideally, include the proper notation for all proteins/genes referred to here-UPPER case italics for human genes, 1st letter upper case followed by Lower case for mice.
- Different fonts are used in the figures- it not clear if this is intentional?
- It would useful if a comment on the importance of nNOS localization and capillary formation was included.
- Figure 4 in particular is very crowded and a little difficult to follow. Perhaps the use of different shapes to represent some of the componenetswould add clarity.
Author Response
Point-by point response to Reviewer 4:
Comment 1: ‘The authors present a timely, well structured, and comprehensive review on dystrophin and its role in skeletal muscle. Suggested edits: 1. Change the title to reflect that the review primarily concerns the role of dystrophin in skeletal muscle; while mention is made of other tissues and cardiac muscle, this is not covered in detail.
Response: We would like to thank Reviewer 4 for the positive evaluation of our manuscript and constructive criticism. We agree to change the title to better show the focus on skeletal muscle dystrophin. The revised title reads now as follows: ‘The dystrophin node as integrator of cytoskeletal organization, lateral force transmission, fiber stability and cellular signaling in skeletal muscle’.
Comment 2: ‘Use the correct notation for all genes- in the abstract the notation Dmd is appropriate for mouse, and since DMD patients are being referred to, DMD should be used’.
Response: The gene name has been changed accordingly.
Comment 3: ‘See also page 9, 4th paragraph. Ideally, include the proper notation for all proteins/genes referred to here-UPPER case italics for human genes, 1st letter upper case followed by Lower case for mice’.
Response: Gene names have been changed in the revised manuscript, but in this case we tried to use the terms as descriptors (without italics) of individual proteins.
Comment 4: ‘Different fonts are used in the figures- it not clear if this is intentional?’.
Response: This was not intended and appears to have been introduced during conversion of a Macintosh-generated WORD file into the reviewer’s PDF file. We have re-formatted our file and hope that this will solve this issue with fonts.
Comment 5: ‘It would useful if a comment on the importance of nNOS localization and capillary formation was included’.
Response: To address this point, we have added the following statement on relevant new references to the revised manuscript: … The localization and continued functioning of nNOS has a crucial regulatory impact on arteriolar blood flow within skeletal muscles. This is clearly shown by the fact that functional ischemia occurs in connection with reduced nitric oxide levels. Narrowing of blood vessels results in oxygen deficiency which in turn renders muscle fibres more susceptible to metabolic stress and cellular degeneration in muscular dystrophy [137-139]. Stabilization of nNOS enzyme at the sarcolemma depends on the structural and functional integrity of …’.
[137] Sato, K.; Yokota, T.; Ichioka, S.; Shibata, M.; Takeda, S. Vasodilation of intramuscular arterioles under shear stress in dystrophin-deficient skeletal muscle is impaired through decreased nNOS expression. Acta Myol. 2008, 27, 30-36.
[138] Tidball, J.G.; Wehling-Henricks, M. Nitric oxide synthase deficiency and the pathophysiology of muscular dystrophy. J. Physiol. 2014, 592, 4627-4638.
[139] Nichols, B.; Takeda, S.; Yokota, T. Nonmechanical Roles of Dystrophin and Associated Proteins in Exercise, Neuromuscular Junctions, and Brains. Brain Sci. 2015, 5, 275-298.
Comment 6: ‘Figure 4 in particular is very crowded and a little difficult to follow. Perhaps the use of different shapes to represent some of the componenets would add clarity’.
Response: In response to another review (by Reviewer 1), Figure 4 was removed from its original position and moved to the lower panel of Figure 1. We have tried in this revised figure to include a model of the full-length dystrophin molecule, as well as all the key players of the core dystrophin complex using different colours and shapes of individual groups of molecules. The colour code was also used in the same way as in revised Figure 3, as suggested by other reviewers. We hope that the revised Figure 1 now gives already at the start of the manuscript a sufficient overview of the dystrophin complex and that this image is sufficiently organized for an informatic presentation of the spatial configuration of the many constituents of the dystrophin complex.
Round 2
Reviewer 1 Report
The authors have assimilated all of our suggestions and have meticulously made great changes, which I believe will make the reading of this review much more accessible to a wider readership. Congratulations!
Reviewer 2 Report
The authors have addressed all points raised, and now the manuscript appears improved.
Reviewer 3 Report
No further comments, the review is improved.